# Rapid Profiling of Soybean Aromatic Compounds Using Electronic Nose

**DOI:** 10.3390/bios9020066

**Published:** 2019-05-24

**Authors:** Ramasamy Ravi, Ali Taheri, Durga Khandekar, Reneth Millas

**Affiliations:** Department of Agricultural and Environmental Sciences, Tennessee State University, 3500 John A, Merritt Blvd, Nashville, TN 37209-1561, USA; rravi@tnstate.edu (R.R.); shama.khandekar@gmail.com (D.K.); renz_millas@yahoo.com (R.M.)

**Keywords:** e-nose, soybean, volatile, beany, electronic nose

## Abstract

Soybean (*Glycine max* (L.)) is the world’s most important seed legume, which contributes to 25% of global edible oil, and about two-thirds of the world’s protein concentrate for livestock feeding. One of the factors that limit soybean’s utilization as a major source of protein for humans is its characteristic soy flavor. This off-flavor can be attributed to the presence of various chemicals such as phenols, aldehydes, ketones, furans, alcohols, and amines. In addition, these flavor compounds interact with protein and cause the formation of new off-flavors. Hence, studying the chemical profile of soybean seeds is an important step in understanding how different chemical classes interact and contribute to the overall flavor profile of the crop. In our study, we utilized the HERCALES Fast Gas Chromatography (GC) electronic nose for identification and characterization of different volatile compounds in five high-yielding soybean varieties, and studied their association with off-flavors. With aroma profiling and chemical characterization, we aim to determine the quantity and quality of volatile compounds in these soybean varieties and understand their effect on the flavor profiles. The study could help to understand soybean flavor characteristics, which in turn could increase soybean use and enhance profitability.

## 1. Introduction

Soybean (*Glycine max* (L.)) seed protein content is 35–50% of its total dry weight and is a major source of protein in the human diet and for animal nutrition. Soybean protein also has a well-balanced amino acid profile and is rich in many essential amino acids. Soybean meal has been used extensively to make popular food products such as tofu, soy milk, soybean paste (miso), green soybeans (edamame), boiled beans (nimame), fermented soybeans (natto), soy sauce (shoyu), soybean sprouts (moyashi), and roasted soybean flour (kinako). Soybean consumption has been limited in the western world due to the beany flavor present in soy meal products. Enzymatic oxidation of linoleic acid and linolenic acid by lipoxygenase genes (Lox) is reported as a major cause of the beany flavor [1,2], and in soybeans there are three separate genes, Lox1, Lox2 and Lox3 controlling this trait [2]. Hexanal is commonly associated with the grassy flavor; hexanol; 1-octen-3-ol; 1-octen-3-one; *trans*,*trans*-2,4-decadienal; and *trans*,*trans*-2,4-nonadienal are other aromatic compounds linked with the beany taste in soy meal products [3]. Odor compounds of soybean products depend on the soybean cultivars and can change in each variety depending on growing season, storage conditions and processing technologies. Boiling the seeds at 100 °C deactivates the lipoxygenase enzymes and is the common method used for reducing the beany flavor. Breeding soybean lines with reduced beany flavor is another approach that can be used for minimizing the off-flavors in soybeans. In order to establish such breeding programs, establishing a reliable and fast screening method for testing beany flavor is necessary. Plant volatile compounds such as beany flavor are overlooked in plant phenotyping. Novel molecular techniques and marker assisted breeding can be used to map the QTLs controlling these traits and find the loci/genetic mechanisms regulating these compounds which can then be exploited for developing soybean lines with reduced off-flavor traits. 

Various volatile compounds may serve as indicators of developmental maturity and as biochemical markers to evaluate seed quality. Several compound classes identified were alcohols, aldehydes, esters and lactones, ketones, and terpenoids. Many reports are available on the key volatiles of soybean [4,5,6,7,8]. The development of objectionable off-odor detection and classification methodology for use in grain grading has stimulated research on volatile components of soybeans and grains [9]. 

## 2. Current Analytical Approaches in Volatile Compound Measurements, Especially in Seeds

Numerous analytical approaches have been developed for measuring volatile compounds and gas exchange measurements in seed samples. Stephen et al. [10] analyzed the soybean seed volatiles using a solid phase microextraction (SPME) method combined with gas chromatography–mass spectrometry (GC-MS) and reported that 30 known volatile compounds were recovered, and that an additional 19 new compounds were identified, or tentatively identified. During early periods of development at maturity stage R6, several volatiles were present at relatively high concentrations, including 3-hexanone, (E)-2-hexenal, 1-hexanol, and 3-octanone. At maturity stage R7 and R8, decreased amounts of 3-hexanone, (E)-2-hexenal, 1-hexanol, and 3-octanone were observed. At maturity stage R8, hexanal, (E)-2-heptenal, (E)-2-octenal, ethanol, 1-hexanol, and 1-octen-3-ol were detected at relatively high concentrations. 

An investigation by Shu et al. [11] using an aroma extract dilution analysis (AEDA) of the aroma concentrate of soy milk made from a major Japanese soybean cultivar, Fukuyutaka (FK), revealed 20 key aroma compounds having flavor dilution (FD) factors of not less than 64. Among them; 2-isopropyl-3-methoxypyrazine; *cis*-4,5-epoxy-(E)-2-decenal; *trans*-4,5-epoxy-(E)-2-decenal; 3-hydroxy-4,5-dimethyl-2(5H)-furanone; and 2′-aminoacetophenone were identified as the key aroma compounds in soy milk for the first time. Generally, it is believed that aroma compounds might be generated from lipids, amino acids, sugars, and ferulic acid present in food.

Sample preparation and scalability of GC-MS and similar instruments resulted in the development of cheaper, faster, and more user-friendly measurement instruments for routine use in analytical applications. Electronic nose (e-nose) devices are developed as versatile and low-cost alternatives to GC-MS instruments that minimize sample preparation and extraction, and offer many potential uses in biomedical and agriculture applications [12]. Volatile compounds can be measured from the sample headspace with minimal sample preparation time. The objective of this study was to use an e-nose instrument in measuring the volatile compounds among five different soybean cultivars and evaluate its potential as an alternative to GC-MS approach, and as a rapid screening tool for aromatic variations in soybean seeds.

## 3. Materials and Methods

### 3.1. Plant Materials

Five recent soybean releases were selected for these experiments including, UA5014C, UA5414RR, JTN-5503, JTN-5110, and JTN-5203. These lines were reported to have a higher yielding potential in a statewide comparison and resistance to common diseases in southern states of US (Table 1). The UA5014C and UA5414RR lines were developed by the Arkansas Agricultural Experiment Station, while JTN-5503, JTN-5110, and JTN-5203 were developed at USDA-ARS Jackson Research Station. Parental information for these lines is provided in Table 1. These lines were grown at the Tennessee State University research farm in 2017. The experimental unit consisted of three replicates with two rows (20 feet deep) and a planting density of 5 seeds/ft. 

### 3.2. Electronic Nose

The HERCALES GC Flash electronic nose (AlphaMos, Toulouse, France) was used to discriminate the odor patterns of different aroma models. For each variety, 20 gm of seed was weighed and grinded in a grinder (Waring WSG60 Grinder) at high speed for 2 min. The resulting soy flour was weighed (6 gm) and placed in a 20 mL glass vial. Following this, 7 mL of sterile distilled water was added to each tube. The sample was prepared in a septa-sealed screw cap vial and equilibrated for 200 s at 50 °C, separately. Subsequently, the aroma headspace above the sample was introduced into the electronic nose at the speed of 270 μL/s using automatic headspace sampler (PerkinElmer, MA, USA). The column temperature program used for the experiment was 40 °C (1 min)-2 °C/min-200 °C (3 min), and the injection temperature of the injector and detector were set at 180 °C and 220 °C, respectively. In addition, at the end of each column a FID detector was placed and the acquired signal was digitalized every 0.01 s. The Heracles electronic nose is equipped with two columns working in parallel mode. A non-polar column (MXT5: 5% diphenyl, 95% methylpolysiloxane, 10 m length, and 180 μm diameter), and a slightly polar column (MXT1701: 14% cyanopropylphenyl, 86% methylpolysiloxane, 10 m length, and 180 μm diameter). A single comprehensive chromatogram was generated by joining the chromatograms obtained with the two columns. This approach helps reduce incorrect identifications due to overlapping of chromatograms obtained with two different columns, and represents a useful tool for improved identification. For calibration of the instrument, an alkane solution (from n-hexane to n-hexadecane) was used to convert retention time in Kovats indices and to identify the volatile compounds using specific software (AromaChemBase). Each analysis was repeated a total of three times, and all of the response data was analyzed using Alpha Soft software (Version 3.0.0, Toulouse, France). 

### 3.3. Results and Discussion

The volatile profiles were generated using the e-nose and were subjected to PCA analysis. The PCA plot (Figure 1) shows the distinct clusters formed for different soy varieties indicating that the volatile profiles of soy varieties are distinctly different from each other. It also demonstrates the potential use of this system in rapid profiling of volatile compounds in different soybean cultivars. UA5414RR and UA5014C were comparable in their volatile profiles while other samples namely JTN5203, JTN5503, and JTN5110 were distantly diverse different from one another. The different clusters formed for different samples are due to their differential volatile compounds and their composition.

More than 90% of the volatile compounds were identified with Kovats index and Arochembase software in UA5014C (Figure 2). The total volatile composition is distributed between acids, aldehydes, alcohols, esters, pyrazines (Table 2 and Figure 3). However, the major volatile composition was contributed by Ethyl-2-Methyl Butyrate (22.72%), 2-Methyl Propanal (18.21%) and 2-Propanol (16.45%). These three volatile components nearly contribute 50% of the total volatile composition in this cultivar. In UA5414RR (Table 3, Figure 2 and Figure 3), the contribution of Ethyl-2-Methyl Butyrate (24.07%) and 2-Methyl Propanal (19.42%) is still high but instead of 2-Propanol, contribution of Ethyl 2-Methylbutanoate (16.01%) was higher in the total volatile composition. From Figure 3, it is clear that esters were the major contributor of the volatiles followed by aldehydes and alcohols in both UA5414RR and UA5014C. Acids and monoterpenes were not detected in UA5414RR. Alcohol was significantly higher in UA5014C compared to UA5414RR. 

In JTN5503 (Table 4, Figure 4 and Figure 5), Ethyl formate (48.29) and Ethyl-2-Methyl Butyrate (10.12) presence was higher than other compounds whereas in JTN5110 (Table 5, Figure 4 and Figure 5), Dimethyl sulphide (34.2) and Ethyl-2-Methyl Butyrate (14.39) were higher. In JTN5203 (Table 6, Figure 4), Dimethyl sulphide alone contributed to over 64% of the total volatile composition. A visual comparison of the peaks in Figure 4 clearly indicates the differences between JTN cultivars in peaks 2, 5, and 17. From Figure 5, it is clear that esters were the major contributor of the volatiles followed by aldehydes and ketones in JTN5203, JTN5503 and JTN5110. The sulfur containing compounds were a major volatile contributor in JTN5110, and JTN5203 but not in JTN5503. Furans were detected only in JTN5110 and were absent in the other two varieties. In general, acids, furans and pyrazines were low in all the samples. 

PCA analysis indicated that UA5414RR and UA5014C were comparable in their volatile profiles while other samples namely JTN5203, JTN5503, and JTN5110 were distantly different from each other (Figure 1). Different clusters formed in different samples according to their differential volatile compounds and their compositions (Table 2, Table 3, Table 4, Table 5 and Table 6). It should be noted that beany flavor is caused by a combination of different compounds and assigning specific flavor to a cultivar should be carried out using sensory analysis with a panel of trained evaluators. 

## 4. Conclusions

E-nose has been used in a wide range of applications including odor analysis, quality control in food products, and biomedical applications. This study illustrates the use of e-nose as a versatile analysis tool and alternative method for measuring volatile compounds in soybean seeds with minimal sample preparation time. This approach can be used in a high-throughput phenotyping system and for screening different soybean lines. This system can be used as a rapid screening tool in breeding programs, in the selection of soybean mutants/varieties with different volatile profiles, and also for mapping the QTLs and loci responsible for these traits. This platform can also be used to link the beany flavor to seed volatile compounds, ultimately developing varieties with reduced off-flavor taste and better acceptance by the consumer. 

## Figures and Tables

**Figure 1 biosensors-09-00066-f001:**
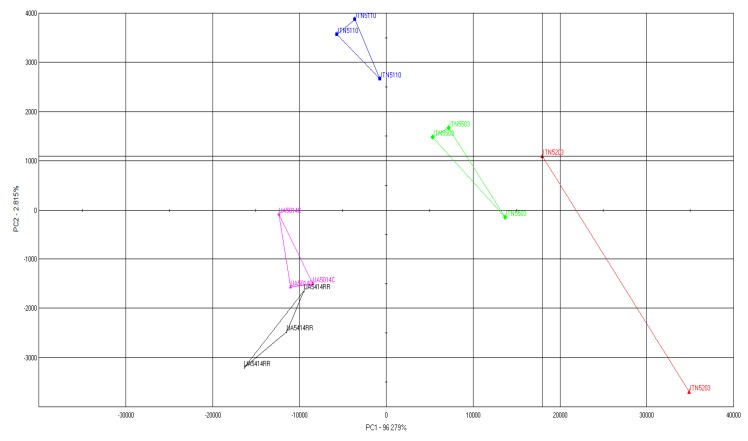
PCA plot of volatile profiles from soybean cultivars.

**Figure 2 biosensors-09-00066-f002:**
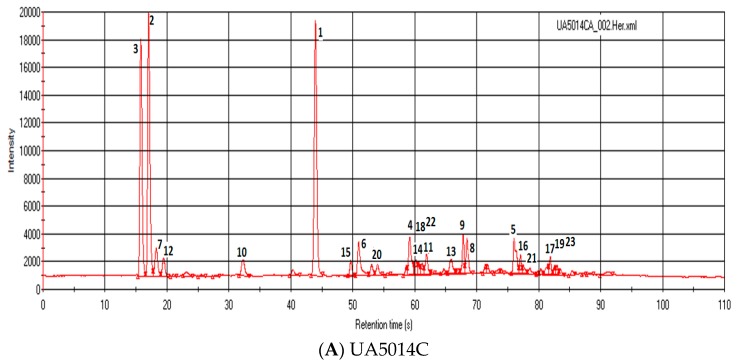
Ultra-fast GC Chromatogram of soybean varieties: UA5014C and UA5414RR.

**Figure 3 biosensors-09-00066-f003:**
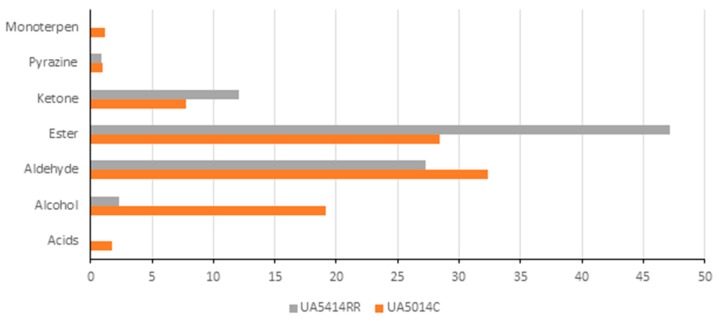
Comparison of volatile compounds in soybean varieties: UA5014C and UA5414RR.

**Figure 4 biosensors-09-00066-f004:**
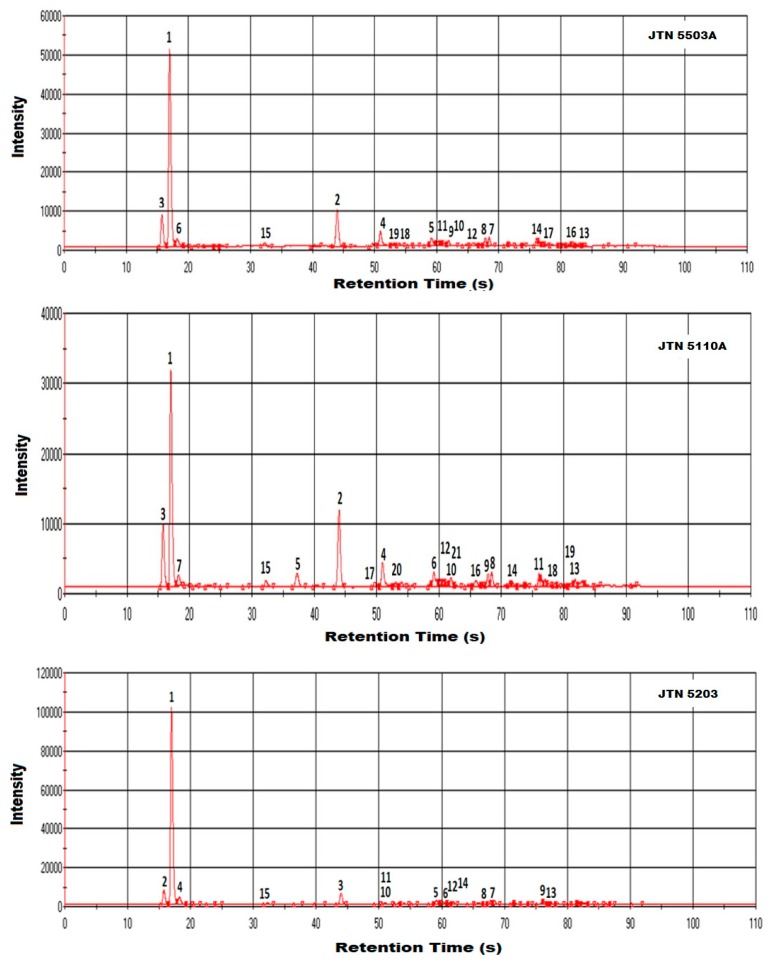
Ultra-fast GC Chromatogram of soybean varieties: JTN5503, JTN5110, JTN5203.

**Figure 5 biosensors-09-00066-f005:**
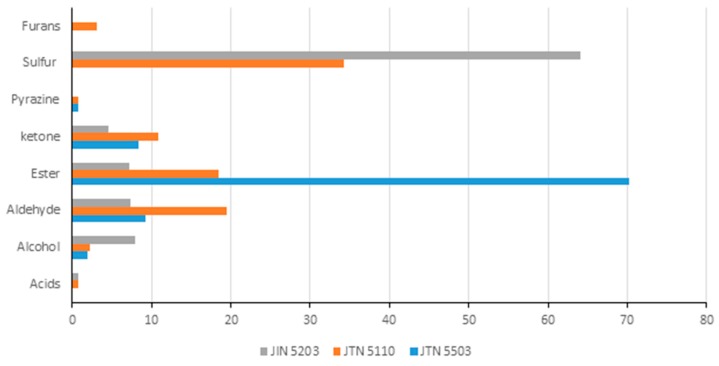
Comparison of volatile compounds in soybean varieties: JTN5503, JTN5110, JTN5203.

**Table 1 biosensors-09-00066-t001:** List of soybean lines, plant introduction (PI) number, pedigree, maturity group (MG) and year of release.

Breeding Material	Plant Introduction #	Parental Lines	MG	Reference
JTN5503	PI 641938	Fowler × Manokin	V (5.5)	Arelli et al. [13]
JTN5110	PI 678369	J98-32 (Manokin × Fowler.) × Anand	V (5)	Arelli et al. [14]
JTN5203	PI 664903	Caviness × Anand	V (5)	Arelli et al. [15]
UA 5014C	PI 675648	Ozark × Anand	V (5)	Chen et al. [16]
UA 5414RR	NA	R96-3427 × 98,601	V (5.4)	Pengyin et al. [17]

**Table 2 biosensors-09-00066-t002:** Headspace volatile compounds of soybean variety: UA5014C.

Peak No	Volatile Compounds				Retention Time (s)	Kovat’s Index
	Name	Surface Percent	Category/Total Percent	Sensory Descriptors		
19	Pentanoic Acid	0.91 ± 0.07	Acids 1.73	Beefy, cheese, pungent, sour, sweaty	81.47	1366
21	Butanoic Acid	0.82 ± 0.06			77.49	1281
3	2-Propanol	16.45 ± 1.26	Alcohols 19.15	Alcoholic, ethereal	15.78	505
12	2-Methyl-1-Propanol	1.54 ± 0.08		Alcoholic, bitter, chemical, glue	19.43	599
15	3-Heptanol	1.16 ± 0.13		Green, herbaceous	49.68	881
2	2-Methyl Propanal	18.21 ± 0.71	Aldehydes 32.35	Burnt, fruity, green malty, pungent, spicy, toasted	17.03	538
4	Benzaldehyde	3.67 ± 0.33		Almond, burnt sugar, fruity, woody	59.17	971
5	P-Anisaldehyde	3.39 ± 0.13		Anise, minty, sweet	76.02	1252
7	Butanal	2.61 ± 0.08		Chocolate, green, malty, pungent	18.26	569
8	N-Nonanal	2.44 ± 0.11		Chlorine, citrus, fatty, floral, fruity, gaseous, gravy, green, lavender	68.42	1110
14	Benzaldehyde	1.28 ± 0.06		Almond, burnt sugar, fruity, woody	60.19	984
22	2-Decenal	0.75 ± 0.01		Fatty, orange	60.67	990
1	Ethyl-2-methyl Butyrate	22.72 ± 1.92	Esters 28.42	Apple, blackberry, fruity, green, strawberry, sweet	43.96	854
9	Ethyl Heptanoate	2.29 ± 0.08		Grape like	67.81	1099
10	Ethyl Butyrate	1.71 ± 0.04		Acetone, banana, bubblegum, caramelized, fruity	32.26	799
11	Hexyl Acetate	1.66 ± 0.04		Acidulous, citrus, fruity, green, herbaceous, sweet wine, tobacco, rubber, spicy	61.90	1007
23	Ethyl Hexanoate	0.04 ± 0.01		Anise, apple, fruity, strawberry, sweet, winegum	82.96	1399
6	2-Heptanone	3.22 ± 0.10	Ketones 7.72	Cheese, cured ham, fruity, gaseous, gravy, nutty, soapy	50.93	887
13	Acetophenone	1.51 ± 0.11		Almond, cheese, floral, musty, sweet	65.87	1069
16	Carvone	1.09 ± 0.02		Minty, warm, herbaceous	77.04	1272
17	Delta Nonalactone	1.05 ± 0.05		Coconut	81.86	1375
20	Gamma Nonalactone	0.85 ± 0.09		Coconut, fruity, peach, woody	53.03	897
18	Trimethyl Pyrazine	0.95 ± 0.08	Pyrazines 0.95	Cocoa, earthy, musty, nutty, peanut, potato, roasted nut	61.20	997
	SUM	90.26 ± 0.08				

**Table 3 biosensors-09-00066-t003:** Headspace volatile compounds of soybean variety: UA5414RR.

Peak No.	Volatile Compounds				Retention Time (s)	Kovat’s Index
	Name	Surface Percent	Category/Total Percent	Sensory Descriptors		
12	3-Heptanol	1.49 ± 0.05	Alcohol 2.29	Green, herbaceous	49.68	881
20	1-Heptanol	0.80 ± 0.04		Green, herbaceous	53.05	897
2	2-Methyl Propanal	19.42 ± 1.59	Aldehydes 27.31	Burnt, fruity, green malty, pungent, spicy, toasted	17.05	538
5	Benzaldehyde	3.15 ± 0.23		Almond, burnt sugar, fruity, woody	59.16	970
7	p-Anisaldehyde	2.78 ± 0.10		Anise, minty, sweet	76.05	1252
9	Phenylmethanal	1.96 ± 0.12		Almond, burnt sugar, fruity, woody	60.20	984
1	Ethyl-2-Methyl Butyrate	24.07 ± 4.36	Esters 47.19	Apple, blackberry, fruity, green, strawberry, sweet	43.98	855
3	Ethyl 2-Methylbutanoate	16.01 ± 3.44		Apple, blackberry, fruity, green, strawberry, sweet	15.80	506
8	Ethyl Heptanoate	2.11 ± 0.10		Grape like	68.43	1110
10	Ethyl Enanthate	1.94 ± 0.10		Acidic, fruity	67.82	1099
11	Hexyl Acetate	1.61 ± 0.12		Acidulous, citrus, fruity, green, herbaceous, sweet wine, tobacco, rubber, spicy	61.92	1007
13	Ethyl Butyrate	1.45 ± 0.08		Acetone, banana, bubblegum, caramelized, fruity	32.30	799
4	2-Heptanone	4.30 ± 0.30	Ketones 12.07	Cheese, cured ham, fruity, gaseous, gravy, nutty, soapy	50.95	887
6	Butane-2,3-Dione	2.83 ± 0.12		Butter, caramelized, creamy, fruity, pineapple, spirit	18.28	570
14	Acetophenone	1.35 ± 0.03		Almond, cheese, floral, musty, sweet	65.92	1069
16	Delta Nonalactone	1.04 ± 0.09		Coconut	81.96	1377
17	Carvone	0.96 ± 0.01		Minty, warm, herbaceous	77.08	1273
19	Delta Nonalactone	0.86 ± 0.03		Coconut	83.08	1401
21	γ-Nonalactone	0.73 ± 0.01		Coconut, fruity, peach, woody	81.57	1368
15	ß-Pinene	1.21 ± 0.06	Monoterpens 1.21	Terpenic	60.68	990
18	2,5-Dimethyl Pyrazine	0.89 ± 0.02	Pyrazines 0.89	Chocolate, cocoa, medicinal, roast beef, roasted nut, woody	54.03	905
	SUM	90.94				

**Table 4 biosensors-09-00066-t004:** Headspace volatile compounds of soybean variety: JTN5503.

Peak No.	Volatile Compounds				Retention Time (s)	Kovat’s Index
	Name	Surface Percent	Category/Total Percent	Sensory Descriptors		
13	Methyl Eugenol	1.15 ± 0.07	Alcohol 1.9	Clove, spicy.	83.00	1400
19	3-Heptanol	0.75 ± 0.04		Green, herbaceous	53.04	897
5	Benzaldehyde	2.51 ± 0.11	Aldehyde 9.3	Almond, burnt sugar, fruity, woody	59.15	970
6	Butanal	2.43 ± 0.14		Chocolate, green, malty, pungent	18.21	568
7	N-Nonanal	1.86 ± 0.16		Chlorine, citrus, fatty, floral, fruity, gaseous, gravy, green, lavender	68.44	1110
10	P-Anisaldehyde	1.28 ± 0.08		Anise, minty, sweet	76.06	1253
11	Benzaldehyde	1.22 ± 0.07		Almond, burnt sugar, fruity, woody	60.19	984
1	Ethyl Formate	48.29± 3.85	Ester 70.21	Smell of rum, Ethereal, pungent	16.98	536
2	Ethyl-2-Methyl Butyrate	10.12± 1.14		Apple, blackberry, fruity, green, strawberry, sweet	43.97	855
3	Methyl Formate	7.76 ± 0.58		Ethereal, pungent	15.75	505
8	Ethyl Heptanoate	1.60 ± 0.12		Grape like	67.83	1099
9	Hexyl Acetate	1.35 ± 0.06		Acidulous, citrus, fruity, green, herbaceous, sweet wine, tobacco, rubber, spicy	61.91	1007
15	Ethyl Butyrate	1.09 ± 0.06		Acetone, banana, bubblegum, caramelized, fruity	32.26	799
4	2-Heptanone	4.38 ± 0.63	Ketone 8.41	Cheese, cured ham, fruity, gaseous, gravy, nutty, soapy	50.93	887
12	Acetophenone	1.19 ± 0.05		Almond, cheese, floral, musty, sweet	65.91	1069
14	(+)-Carvone	1.13 ± 0.06		Caraway, minty, peppermint	76.38	1259
16	Gamma Nonalactone	0.87 ± 0.03		Coconut, fruity, peach, woody	81.88	1375
17	(−)-Carvone	0.84 ± 0.02		Caraway, minty, peppermint	77.09	1273
18	2,5-Dimethyl Pyrazine	0.74 ± 0.09	Pyrazines 0.74	Chocolate, cocoa, medicinal, roast beef, roasted nut, woody	53.99	904
	SUM					

**Table 5 biosensors-09-00066-t005:** Headspace volatile compounds of soybean variety: JTN5110.

Peak No	Volatile Compounds				Retention Time (s)	Kovat’s Index
	Name	Surface Percentage	Category/Total Percent	Sensory Descriptors		
20	Pentanoic Acid	0.83 ± 0.05	Acids 0.83	Beefy, cheese, pungent, sour, sweaty	53.99	904
17	3-Heptanol	0.95 ± 0.03	Alcohol 2.29	Green, herbaceous	46.69	881
13	Methyl Eugenol	1.34 ± 0.07		Clove, spicy	82.95	1398
18	2-Decenal	0.92 ± 0.05	Aldehyde 19.5	Fatty, orange	77.04	1272
6	Benzaldehyde	2.94 ± 0.19		Almond, burnt sugar, fruity, woody	59.17	971
12	Phenylmethanal	1.39 ± 0.08		Burnt sugar, fruity, woody	60.19	984
7	Butanal	2.45 ± 0.14		Chocolate, green, malty, pungent	18.25	569
11	P-Anisaldehyde	1.45 ± 0.09		Anise, minty, sweet	76.02	1252
3	Propanal	10.35 ± 0.33		Ethereal, plastic, pungent, solvent	15.76	505
15	Ethyl Butyrate	1.28 ± 0.06	Ester 18.5	Acetone, banana, bubblegum, caramelized, fruity	32.29	799
16	Mthyl Butyrate	1.25 ± 0.10		Banana, bubblegum, caramelized, fruity	65.92	1069
2	Ethyl-2-Methyl Butyrate	14.39 ± 1.63		Apple, blackberry, fruity, green, strawberry, sweet	43.98	855
10	Hexyl Acetate	1.58 ± 0.09		Acidulous, citrus, fruity, green, herbaceous, sweet wine, tobacco, rubber, spicy	61.92	1007
5	Furfural	3.12 ± 0.40	Furans 3.12	Almond, bread, sweet	37.26	823
14	(-)-Carvone	1.30 ±0.06	Ketone 10.86	Caraway, minty, peppermint	76.35	1258
4	2-Heptanone	4.80 ± 0.36		Cheese, cured ham, fruity, gaseous, gravy, nutty, soapy	50.95	887
19	Delta Nonalactone	0.88 ± 0.06		Coconut	81.84	1374
8	Ethyl Heptanoate	2.09 ± 0.12		Grape like	68.43	1110
9	Ethyl enanthate	1.79 ± 0.12		Pleasant, floral	97.84	1099
21	Trimethyl Pyrazine	0.80 ± 0.03	Pyrazine 0.8	Cocoa, earthy, musty, nutty, peanut, potato, roasted nut	60.67	990
1	Dimethyl Sulphide	34.20 ± 3.31	Sulfur 34.2	Cabbage, fruity, gaseous, gasoline, moldy, vegetable soup	17.01	537
	SUM	90.09 ± 0.15				

**Table 6 biosensors-09-00066-t006:** Headspace volatile compounds of soybean variety: JTN 5203.

Peak No.	Volatile Compounds	Surface Percentage	Category	Sensory Descriptors	Retention Time (s)	Kovat’s Index
15	Butanoic Acid	0.82 ± 0.02	Acids 0.82	Butter, cheese, rancid, sweaty	65.87	1069
2	2-Propanol	5.84 ± 0.63	Alcohol 7.93	Alcoholic, ethereal	15.57	505
10	3-Heptanol	1.04 ± 0.16	3-Heptanol	75.96	1251
13	Methyl Eugenol	1.05 ± 0.02	Clove, spicy	76.29	1257
5	Benzaldehyde	2.13 ± 0.41	Aldehyde 7.28	Almond, burnt sugar, fruity, woody	18.23	568
6	Phenylmethanal	1.46 ± 0.15	Burnt sugar, fruity, woody	59.12	970
7	N-Nonanal	1.37 ± 0.10	Chlorine, citrus, fatty, floral, fruity, gaseous, gravy, green, lavender	60.16	983
8	Nonanaldehyde	1.21 ± 0.15	Chlorine, citrus, fatty, floral, fruity, gaseous, gravy, green, lavender	68.38	1109
9	p-Anisaldehyde	1.11 ± 0.22	Anise, minty, sweet	67.77	1098
3	Ethyl-2-Methyl Butyrate	5.26 ± 0.70	Ester 7.24	Apple, blackberry, fruity, green, strawberry, sweet	48.81	889
11	2,3-Hexen-1-Ol, Acetate	1.01 ± 0.13	Banana, fruity, green, sweet, sharp	50.95	887
12	Phenyl Ethyl Acetate	0.97 ± 0.16	Phenyl Ethyl Acetate	61.86	1006
4	Butane-2,3-Dione	3.52 ± 0.37	Ketone 4.5	Butter, caramelized, creamy, fruity, pineapple, spirit	43.97	855
14	Acetophenone	0.98±0.01	Almond, cheese, floral, musty, sweet	82.86	1397
1	Dimethyl Sulphide	64.14 ± 3.91	Sulfur 64.14	Cabbage, fruity, gaseous, gasoline, moldy, vegetable soup	16.97	536
	Sum	91.92 ± 0.18

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
