# Peer review of "Rapid Profiling of Soybean Aromatic Compounds Using Electronic Nose"

_biosensors, 2019, doi:10.3390/bios9020066_

Reviewer 1 Report

As presented, the study is just a report concerning the volatile compounds found in the 5 selected soybean cultivars. The methods used and the apparatus are appropriate for the study, but this type of profiling could have been done with any other GC with a single column, not necessarily an electronic nose with two parallel columns. The only advantage of using the present equipment is the rapidity of the compound identification and lower cost, but not the main feature for which the instrument was designed – namely, to group samples in accordance to certain volatile profiles and correlate them with sensory traits.

While the volatile profiles of the 5 soybean cultivars were determined and reported in detail, no attempt was made to find correlations of the profiles with any sensory parameters, to point out which cultivar/group of cultivars present a beany flavor, a trait considered less acceptable for the consumers and supposedly a cause of other off-flavors. The separation performed by the electronic nose of the two groups of cultivars can be interesting, but only if some sensory traits, especially a certain degree of beany flavor, can be correlated with the overall aromatic profile of these groups. The PCA separation based on the volatile profile only shows that two cultivars have similar profiles, while the rest are distinct, an information which is of limited practical use without the sensory description of the cultivars.

An attempt was made to introduce in discussion the beany flavor (lines 177-193), but this sub-chapter was only a review of the some knowledge found in 2 other studies, with little relevance for the present cultivars or the present method. This sub-chapter would rather belong in “Introduction”, not in “Results and discussion”. One mentioned study found two compounds likely to lead to beany flavor, but none of those were identified in the 5 cultivars evaluated this time. The other study showed that several volatile compounds from soybeans have flavors unacceptable to consumers, but the authors do not try to link these findings with their results. 

The authors presented the compounds found in each cultivar and added sensory descriptors for each individual compound (Tables 2 and 3). However, none of the compounds alone leads to a specific beany flavor. It rarely happens that a single certain compound gives a fruit its major flavor/aroma; usually, the overall combination of certain volatile compounds determines the specific flavor. This can only be defined by using sensory analysis with a panel of evaluators. Only after that, based on the correlation with volatile profiles, the electronic nose could be used to screen for cultivars with a certain desirable/undesirable sensory trait.

Without clear identification of the volatile profiles which lead to the undesirable beany flavor, so that the selection programs may target those traits, the present paper is of limited importance as a screening tool in breeding programs or for “mapping the QTLs and loci responsible for these traits”. Such bold conclusions are not supported by the data presented in this study.

Valuable information was presented regarding the volatile chemical compounds of the 5 cultivars and this can be useful for future research on those particular cultivars. However, to be meaningful for more than these 5 specific cultivars, the paper should include sensory analysis of the soybean extracts and a correlation with the volatile profiles.

Minor comment:

Lines 63-64 should be express differently,  as to many “reported” are found in the same phrase. 

Author Response

1.       As presented, the study is just a report concerning the volatile compounds found in the 5 selected soybean cultivars. The methods used and the apparatus are appropriate for the study, but this type of profiling could have been done with any other GC with a single column, not necessarily an electronic nose with two parallel columns. The only advantage of using the present equipment is the rapidity of the compound identification and lower cost, but not the main feature for which the instrument was designed – namely, to group samples in accordance to certain volatile profiles and correlate them with sensory traits.

Response: Thank you very much for the input. Using GC is the routine approach in profiling volatile compounds but as the reviewer has also mentioned it requires sample prep and extraction which delays rapid profiling of multiple plant samples and populations. We have addressed this point in the new version of the manuscript in introduction (Lines 80 and 84) and discussion section (lines 2019-215). Another advantage of using the E-nose is that it provides the comparison / discrimination based on the global volatile analysis. This sample to sample comparison cannot be done with another equipment.

2.       While the volatile profiles of the 5 soybean cultivars were determined and reported in detail, no attempt was made to find correlations of the profiles with any sensory parameters, to point out which cultivar/group of cultivars present a beany flavor, a trait considered less acceptable for the consumers and supposedly a cause of other off-flavors. The separation performed by the electronic nose of the two groups of cultivars can be interesting, but only if some sensory traits, especially a certain degree of beany flavor, can be correlated with the overall aromatic profile of these groups. The PCA separation based on the volatile profile only shows that two cultivars have similar profiles, while the rest are distinct, an information which is of limited practical use without the sensory description of the cultivars.

Response: We agree with the valuable suggestions by the reviewer as linking the profile with the sensory parameters makes the manuscript more valuable. However, since the graduate student working on the project has already finished her project and training a sensory panel for the beany taste takes time, we won’t be able to accomplish this task in a 10days time limit as requested by MDPI. We will start working on the sensory aspect of the project after recruiting new personnel for this project and hopefully we will be able to provide the sensory panel report in a later publication. We have addressed this point in the discussion part of the manuscript (Lines 186-188). However, it is worth to mention here that based on the Arochembase library all the individual volatile peaks were also characterized with their associated sensory descriptors- which is reported and discussed in the manuscript.

3.       An attempt was made to introduce in discussion the beany flavor (lines 177-193), but this sub-chapter was only a review of the some knowledge found in 2 other studies, with little relevance for the present cultivars or the present method. This sub-chapter would rather belong in “Introduction”, not in “Results and discussion”. One mentioned study found two compounds likely to lead to beany flavor, but none of those were identified in the 5 cultivars evaluated this time. The other study showed that several volatile compounds from soybeans have flavors unacceptable to consumers, but the authors do not try to link these findings with their results. 

Response: As per the suggestion, We have removed this section from the manuscript.

4.       The authors presented the compounds found in each cultivar and added sensory descriptors for each individual compound (Tables 2 and 3). However, none of the compounds alone leads to a specific beany flavor. It rarely happens that a single certain compound gives a fruit its major flavor/aroma; usually, the overall combination of certain volatile compounds determines the specific flavor. This can only be defined by using sensory analysis with a panel of evaluators. Only after that, based on the correlation with volatile profiles, the electronic nose could be used to screen for cultivars with a certain desirable/undesirable sensory trait.

We really appreciate this input and have mentioned this valid point in the discussion (Lines 186-188). As it is clear from the introduction, our major focus was on the characterization on the volatiles and the beany flavor volatile may or may not present in the samples, which depends on many factors, particularly variety, genetic factors, pre-cursors, weather, environmental conditions etc.

5.       Without clear identification of the volatile profiles which lead to the undesirable beany flavor, so that the selection programs may target those traits, the present paper is of limited importance as a screening tool in breeding programs or for “mapping the QTLs and loci responsible for these traits”. Such bold conclusions are not supported by the data presented in this study.

Response: We appreciate the reviewers input. However, this approach is not just focusing on beany flavor, and other volatile compounds which might not be involved in beany flavor can be profiled using this approach as long as they differ between different cultivars. These variations can be screened in crossing different parental lines and better understanding of genetic mechanism controlling these traits in mapping populations.

6.       Valuable information was presented regarding the volatile chemical compounds of the 5 cultivars and this can be useful for future research on those particular cultivars. However, to be meaningful for more than these 5 specific cultivars, the paper should include sensory analysis of the soybean extracts and a correlation with the volatile profiles.

Response: We agree with the valuable suggestion by the reviewer as linking the sensory analysis with the volatile compounds the manuscript more valuable. However, since the graduate student working on the project has already finished her project, we won’t be able to accomplish this task in a 10days time as requested by MDPI. We will start working on the sensory aspect of the project after recruiting new personnel for this project and hopefully we will be able to provide the sensory panel report in a later publication. We have addressed this point in the discussion part of the manuscript (Lines 177-179).

7.       Lines 63-64 should be express differently,  as to many “reported” are found in the same phrase.

Response: These sentences has been revised completely (lines 63-64).

Reviewer 2 Report

A minor revision is needed to make this manuscript suitable for publication.

Lines 177-193: This section, “Beany flavors”, should be deleted because the authors did not study the aroma-active compounds of the soybean by AEDA in this study.

Author Response

1.       Lines 177-193: This section, “Beany flavors”, should be deleted because the authors did not study the aroma-active compounds of the soybean by AEDA in this study.

Response: As per the suggestion, We have removed this section from the manuscript.

Reviewer 3 Report

The results of this article are not described comprehensively in the discussion of data analysis

Author Response

1.       The results of this article are not described comprehensively in the discussion of data analysis

Response: Thanks for the suggestion. We have revised the discussing section, removed the beany flavors and added the reference to the figures and tables in the discussion.

Round  2

Reviewer 1 Report

The main suggestions were taken into account by the authors.